# Using Multiplexed CRISPR/Cas9 for Suppression of Cotton Leaf Curl Virus

**DOI:** 10.3390/ijms222212543

**Published:** 2021-11-21

**Authors:** Barkha Binyameen, Zulqurnain Khan, Sultan Habibullah Khan, Aftab Ahmad, Nayla Munawar, Muhammad Salman Mubarik, Hasan Riaz, Zulfiqar Ali, Asif Ali Khan, Alaa T. Qusmani, Kamel A. Abd-Elsalam, Sameer H. Qari

**Affiliations:** 1Institute of Plant Breeding and Biotechnology, MNS University of Agriculture Multan, Old Shujabad Road, Multan 60000, Pakistan; barkha.binyameen777@gmail.com (B.B.); zulfiqar.ali@mnsuam.edu.pk (Z.A.); asifpbg@gmail.com (A.A.K.); 2Cotton Biotechnology Lab, Center for Advanced Studies in Agriculture and Food Security, University of Agriculture, Faisalabad 38040, Pakistan; sultan@uaf.edu.pk (S.H.K.); aftab.ahmad@uaf.edu.pk (A.A.); msmubarik@gmail.com (M.S.M.); 3Centre of Agricultural Biochemistry and Biotechnology, University of Agriculture, Faisalabad 38040, Pakistan; 4Department of Chemistry, College of Sciences, United Arab Emirates University, Al-Ain 15551, United Arab Emirates; naylamunawar@gmail.com; 5Institute of Plant Protection, MNS University of Agriculture Multan, Old Shujabad Road, Multan 60000, Pakistan; hasan.riaz@mnsuam.edu.pk; 6Biology Department, Al-Jumum University College, Umm Al-Qura University, Makkah 21961, Saudi Arabia; atqumsani@uqu.edu.sa; 7Plant Pathology Research Institute, Agricultural Research Center (ARC), 9-Gamaa Str., Giza 12619, Egypt; kamelabdelsalam@gmail.com; 8Department of Biology, Genetics and Molecular Biology Central Laboratory (GMCL), Aljumum University College, Umm Al-Qura University, Makkah 21961, Saudi Arabia

**Keywords:** cotton, CLCuD, CRISPR/Cas, *N. benthamiana*, virus inhibition, DsRed, transformation

## Abstract

In recent decades, Pakistan has suffered a decline in cotton production due to several factors, including insect pests, cotton leaf curl disease (CLCuD), and multiple abiotic stresses. CLCuD is a highly damaging plant disease that seriously limits cotton production in Pakistan. Recently, genome editing through CRISPR/Cas9 has revolutionized plant biology, especially to develop immunity in plants against viral diseases. Here we demonstrate multiplex CRISPR/Cas-mediated genome editing against CLCuD using transient transformation in *N. benthamiana* plants and cotton seedlings. The genomic sequences of cotton leaf curl viruses (CLCuVs) were obtained from NCBI and the guide RNA (gRNA) were designed to target three regions in the viral genome using CRISPR MultiTargeter. The gRNAs were cloned in pHSE401/pKSE401 containing Cas9 and confirmed through colony PCR, restriction analysis, and sequencing. Confirmed constructs were moved into Agrobacterium and subsequently used for transformation. Agroinfilteration in *N. benthamiana* revealed delayed symptoms (3–5 days) with improved resistance against CLCuD. In addition, viral titer was also low (20–40%) in infected plants co-infiltrated with Cas9-gRNA, compared to control plants (infected with virus only). Similar results were obtained in cotton seedlings. The results of transient expression in *N. benthamiana* and cotton seedlings demonstrate the potential of multiplex CRISPR/Cas to develop resistance against CLCuD. Five transgenic plants developed from three experiments showed resistance (60−70%) to CLCuV, out of which two were selected best during evaluation and screening. The technology will help breeding CLCuD-resistant cotton varieties for sustainable cotton production.

## 1. Introduction

Cotton (*G. hirsutum* L.) has a tremendous range of applications, from livestock feed to edible oil, textile industry and medical bandages. It is a natural fiber crop and the foundation of one of the world’s biggest textile industries, with an economic impact of approx. 600 billion USD worldwide [1], comprising about half of the global textile market [2]. It is known as “White Gold” and is cultivated in more than 80 countries with an annual total production of about 123.6 million bales [3]. Pakistan is among the top five cotton-producing countries in the world [4]. In 2018, Pakistan produced 9.861 million bales of cotton; however, this was 17.1% less than production in 2016 [5]. Multiple factors were responsible for the decline in cultivation and production of cotton in Pakistan; these included market issues, hefty losses caused by a severe attack from pink bollworm, and outbreaks of sucking insects, particularly whitefly. Whitefly is a vector for the cotton leaf curl virus (CLCuV), which causes cotton leaf curl disease (CLCuD). CLCuD is the most destructive disease of cotton in Pakistan and India. Infected cotton plants display explicit symptoms such as vein-thickening, curling of the leaves upwards or downwards, hindered development, and growth of leaf-like enations [6]. CLCuV, which belongs to the *Geminiviridae* family of plant viruses, is responsible for serious impacts on cotton yield and fiber quality production in Pakistan [7,8]. Whitefly-transmitted geminiviruses, the begomoviruses, are a significant group of plant viruses that cause heavy economic loss in the crops [9]. As well as serving as a vector for CLCuV, the whitefly also harms the plants by its feeding [10]. The cotton species *G. hirsutum* is the most broadly cultivated, but it is susceptible to begomoviruses which seriously limit fiber yield and quality [11]. Genome editing has transformed the agricultural landscape with many success stories of engineering biotic stress tolerance in crop plants. The tremendous potential of genome editing to improve crop plants has been demonstrated using model systems for various traits such as biotic and abiotic stress tolerance, nutritional enhancement, and yield improvement [12]. In prokaryotes, CRISPR/Cas9 was originally discovered as a natural defense mechanism against viral attacks [13]. CRISPR/Cas has been proved effective for the targeting and redesigning of crop genomes, especially crops such as cotton which are difficult to manipulate with conventional genetic engineering [14]. Cas9 nuclease has been characterized with two catalytic domains (HNH and RuvC) for creating double-strand breaks (DSBs) in the targeted genomic site. The specificity of CRISPR/Cas9 system to create DSBs is driven by two RNA molecules; trans-activating crRNA (tracrRNA) and CRISPR RNA (crRNA)—and a protospacer adjacent motif (PAM) necessary for recognition of target site in the genome [15]. Although, the CRISPR/Cas9 system has been the most extensively used genome editing system by the scientific community, in recent years, several new CRISPR/Cas variants such as CRISPR/Cas12, CRISPR/CasX, CRISPR/Cas13, CRISPR/Cas3, and CRISPR/Cas14 have been discovered with different PAM requirements, smaller Cas proteins, and improved cleavage efficiency. CRISPR/Cas9, CRISPR/Cas12, and CRISPR/Cas13 have been successfully used for genetic improvement of plants for desired traits [15,16]. CRISPR/Cas9 has also been demonstrated to be effective for viral interference in plants [17,18,19,20,21]. Efforts have been made previously to inhibit viral replication in plants by targeting regulatory sites or sequences with artificial DNA-binding proteins [18,19,22], but no complete resistance has been obtained. The main purpose of this study was to evaluate the efficiency of CRISPR/Cas9 system against CLCuV in *N. benthamiana* as a model system and then in cotton seedlings. We hope the study will be helpful to decrease the economic losses caused by CLCuD in Pakistan.

## 2. Results

### 2.1. Confirmation of gRNAs and sgRNAs in Cas9 Vector through Colony PCR

sgRNA cloning in pHSE401 was first confirmed through colony PCR (Figure 1) with specific primers showing 280-bp amplicons on agarose gel. The colony PCR results showed a significantly high ratio of transformant colonies.

### 2.2. Confirmation of sgRNAs through Restriction Analysis

CRISPR constructs confirmed through digestion with *Hind*III showed around 2000 bp length having multiple gRNAs (Figure 2).

### 2.3. Virus Infectivity Assay in N. benthamiana Plants

*N. benthamiana* plants infiltrated with infectious clones showed severe symptoms of leaf curling in infected and systemic leaves at 14 dpi compared with control plants (Figure 3).

### 2.4. qPCR-Based Determination of Virus Accumulation

qPCR analysis, to determine viral accumulation in control and Cas9-sgRNA infiltrated leaves (Figure 4), revealed that the plants co-infiltrated with Cas9-gRNA and virus infectious clones showed mild symptoms of CLCuV (Figure 5 and Figure 6). A significant decrease in viral symptoms in the range of 60–80% was observed in leaves infiltrated with Cas9-gRNA (Figure 4). In addition, compared to control plants, virus titer was found to be low in systemic leaves infiltrated with Cas9-gRNA, as shown in Figure 6. These results highlight the potential of multiplex genome editing with CRISPR/Cas to control CLCuV in cotton.

For CLCuV infection in cotton, the plants were infiltrated with CLCuV, as shown in Figure 7. The seedlings were also kept in net cages for infection by viruliferous whitefly (Appendix A). Details of the experiment and results of an evaluation of multiplex CRISPR for suppression of CLCuV in *N. benthamiana* and cotton are given in Appendix A. Interference in the viral genome was also confirmed through DNA sequence. All three targets in the viral genome showed disruption at the target site (Appendix A).

### 2.5. Transformed Callus of Cotton

Once a multiplex construct was evaluated in *N. benthamiana*, the same construct was used to transform cotton. In addition, DsRed was transformed in cotton to optimize the transformation protocol. About one-week-old cotton hypocotyls were transformed with Agrobacterium containing multiplex CRISPR/Cas constructs. Transgenic calli were selected on hygromycin/kanamycin; after 4–6 weeks of transformation, cotton callus had started to appear, as shown in Figure 8. The callus was sub-cultured after every four weeks onto freshly prepared plates to avoid contamination. DsRed was transformed as control along with CRISPR/Cas-mediated transformation in cotton.

### 2.6. PCR-Based Confirmation of Transgene

Transformation in cotton callus, embryos, and plantlets was confirmed with PCR using specific primers. The callus was cultured throughout on selection plates using appropriate antibiotics. A specific amplicon of 280 bp was amplified using PCR to indicate a successful transformation of cotton hypocotyls (Figure 9). In addition, DsRed callus was pink to red and visible to the naked eye and under a microscope (Figure 10).

### 2.7. Regeneration of Cotton Callus

Regenerating calli with DsRed expression is shown in Figure 10. Cotton calli with a CRISPR construct were subjected to embryogenesis and transgenic plant development. Based on the number of hypocotyl cuttings used in one Petri plate, 40–50% efficiency in callus induction following 10–20% efficiency for embryogenesis was observed. Expression of gRNAs and Cas9 in the plantlets was confirmed and found successful for virus interference in the screening experiments in *N. benthamiana* and cotton (Figure 11).

Embryos were observed after five to six sub-cultures; five transgenic seedlings were obtained and were subjected to hardening (Figure 12). The details of number of explants developed into plants are given in Appendix A. The transgene was confirmed through PCR as given in Section 2.6, showing similar bands on agarose gel. Expression of gRNAs and Cas9 was observed through RT-PCR (Figure 13).

### 2.8. Evaluation and Screening

All T1 plants were screened for resistance under controlled conditions using viruliferous whitefly (Appendix A). Non-transgenic plants showed complete leaf curling. Compared to control plants, all T1 plants showed 60–70% resistance to CLCuV. Based on resistance evaluation, two transgenic plants were finally selected for developing transgenic lines. The comparison of transformed and control plants for CLCuD resistance has been given in Figure 14. Non-transformed plants showed various symptoms ranging from vein thickening and leaf curling to enations. In comparison, CRISPR plants showed resistance to CLCuD infection when challenged with viruliferous whitefly as seen in Figure 14.

## 3. Discussion

Begomoviruses damage crops each year and undermine food security worldwide [23,24]. Plants exhibiting resistance are the most feasible way to cope with viral diseases transmitted by insects. The conventional techniques of plant breeding have been comprehensively utilized against begomoviruses for developing resistance. However, these techniques are not viable because of the ability of begomoviruses to evolve quickly through recombination and mutation. These challenges require new methodologies able to produce cost-effective resistance as needed. There are two major strategies to use the CRISPR/Cas system against plant viruses. Firstly, targeting viral genomes for degradation and interference to inhibit viral replication and infection. Secondly, targeting host susceptibility factors to enhance plant immunity against viral infection [25]. DNA viruses have been targeted with CRISPR/Cas9 by several researchers [17,18,19,20,22,26,27,28].

CRISPR/Cas9 is a more specific, efficient, and faster GE tool [29]. Applied fruitfully in a variety of organisms, including humans, the technology of CRISPR/Cas9 is constantly under development in genome editing, both in eukaryotes and prokaryotes. Improvements in this technique have made it possible to accomplish high-throughput screening for gene identification at the genomic level [30]. Additional studies show Cas9 modification to prompt transcriptional activation and gene knockdown [31,32]. Applications of CRISPR have also been extended to identify and characterize transcriptional enhancers in cells [31]. Upgrades of this technology have encouraged its use in plants, particularly for modification of major crops such as maize, soybean, wheat, and rice [32,33]. Successful use of the technology in cotton will expand its use in plants, especially polyploidy species. In our research, CRISPR/Cas9 technology was utilized for viral suppression in *N. benthamiana* and cotton. For this purpose, multiple targets in the viral genome were selected, including the nona-nucleotide region, due to its critical importance (Appendix A). Targeting nona-nucleotide, which is important for viral replication, for viral interference and inhibition of replication may produce a broad-spectrum resistance against geminiviruses [20,22]. In infiltration studies, infectious clones were used for virus infection and Cas9 and gRNA expression, which was quantified through RT-PCR from plant leaves. A similar method has been demonstrated by Ji et al. [17], indicating that suitable expression of Cas9, as well as sgRNA, is very important for editing using a CRISPR/Cas-gRNA system. Genome editing with CRISPR/Cas9 is based on RNA-guided DNA-binding of Cas protein to target any sequence in the genome, which may be achieved with higher editing efficiency than with ZFNs and TALENs [29,34,35]. Previous studies showed that partial resistance to begomoviruses may be achieved in *N. benthamiana* by utilizing a TALE protein that targets the conserved motifs in the virus genome [36]. The technology of CRISPR/Cas has been strongly advocated for use against viral diseases [27]. In plant biology, multiplexed editing of a genome has previously revealed wide applications and is known to significantly assist precision breeding and genome engineering in crops. It presents an outstanding opportunity for plant breeding and is being applied to a rising number of plant species owing to its rapidness, precision, and low cost. In particular, it has been effective for the inhibition of DNA viruses in plants [17,20]. However, plant viruses are sometimes immune to small insertions and deletions produced by CRISPR/Cas and consequently can escape after repair of the DSB [17,26].

Practical demonstrations show that the technology of CRISPR/Cas9 could be harnessed for targeting the CLCuV intervention to produce CLCuD resistance. The sgRNA complementary to non-coding and coding CLCuV sequences cloned into Cas9 transgenic cotton plants significantly inhibit the spread of CLCuD, consequently improving disease tolerance. This approach has been demonstrated for suppression of plant DNA viruses [17,18]. Moreover, multiplex genome editing using Cas9 [20] or other CRISPR/Cas variants have been developed for genetic improvement in monocots and dicots [16]. Multiplex genome editing is predicted to be an efficient and effective approach for the development of virus-resistant crops. Simultaneous targeting of multiple sites in viral DNA through CRISPR/Cas can significantly improve virus resistance in plants. This strategy may open novel avenues for engineering the resistance against DNA viruses in other plants. The approach may be exploited further for genome editing of viruses belonging to *Geminivirdae*, hence raising the prospect for achieving sustainable agriculture by controlling diseases caused by DNA viruses.

In the future, improvements on these fronts will bring genome biology into a new era—an era where scientists can edit genomes extensively through a multiplex genome editing strategy. Off-target events are the most important concern for the application of gene editing, particularly with CRISPR/Cas9 [30]. The field of CRISPR technology is also rapidly developing. For example, a base editing tool, which is based on dCas9 combined with a cytidine deaminase, has been reported [37]. In time, we are likely to see more utilization of genome editing platforms to create varieties with broad resistance, which will ultimately assist efforts to satisfy worldwide food demands. Sustainable crops will profit from the coming advancements in genome editing. Researchers who have recently developed resistant lines by genome modification may broaden their work by knockout of negative regulators of beneficial traits to produce the best performing crop. Furthermore, the advancement of multi-trait-enhanced cultivars through numerous knockouts by CRISPR/Cas9 can transform the foundation of plant science. Indeed, we believe that CRISPR, a multiplexed editing tool, may lead in time to the next green revolution.

## 4. Materials and Methods

### 4.1. Sequence Analysis, Selection of Target Site, and gRNA Designing

Sequences of DNA-A of CLCuV were retrieved from the National Center for Biotechnology Information (NCBI) (https://www.ncbi.nlm.nih.gov/ accessed on 13 April 2018) and analyzed for target site selection using ClustalW2 (https://www.ebi.ac.uk/Tools/msa/clustalw2/ accessed on 18 April 2018) and T-Coffee (http://tcoffee.crg.cat/apps/tcoffee/all.html, accessed on 29 September 2021) to find potential sites in consensus sequences for broad-spectrum resistance against a complex of viruses responsible for CLCuD (Appendix A). The single-guide RNAs (sgRNAs) were designed with CRISPRdirect (https://crispr.dbcls.jp/ accessed on 20 April 2018), while CRISPR MultiTargeter (http://multicrispr.net/index.html accessed on 11 May 2018) was used to design the multiple-guide RNAs (multiple gRNAs). The sgRNAs and multiple gRNAs were selected based on high on-target and minimum off-target predictions. Primers were designed according to the target site and synthesized including *Bsa*I restriction sites for cloning.

### 4.2. Cloning of sgRNA and Multiple gRNAs into a Plant Expression Vector (pHSE401/pKSE401) Containing Cas9

For cloning and transient expression of sgRNA along with Cas9, two vectors were used (pHSE401/pKSE401). In final transformation into the plant, pKSE401 Cas9-gRNAs expression vector was used. The details of cloning of sgRNA and multiple gRNAs are given below.

#### 4.2.1. Hybridization of Primers

For hybridization of oligonucleotides, 25 µL (100 µM) of each primer (forward and reverse) were collected in a 200 µL polymerase chain reaction (PCR) tube. Oligo annealing was carried out in a thermocycler using the following program:95 °C (3 min) + 90 °C (1 min) + 80 °C (1 min) + 70 °C (1 min) + 60 °C (1 min) + 50 °C (1 min) + 40 °C (1 min) + 30 °C (1 min) + 20 °C (1 min) + 4 °C (∞)

For cloning multiple gRNAs into Cas9 expression vector (pKSE401), the protocol provided by Xing et al. [38] was followed. Golden Gate cloning was used to clone three gRNAs in the Cas9 plant expression vector.

#### 4.2.2. Construction of Plant Expression Vector Containing Cas9 and sgRNA

The Cas9 plant expression vector (pHSE401) was digested with *BSaI* and the DNA fragment containing Cas9 was eluted using a GeneJET (Thermo Fisher Scientific) kit. Ligation of sgRNA in digested pHSE401 was carried out at 22 °C for 4 h using T4 DNA ligase (New England BioLabs), according to Khan et al. [22]. After four hours of ligation, the ligation product was transformed into chemically competent cells of *E. coli* DH5α and colony PCR was performed with specific primers (U6-26-F and dT4-R) to confirm sgRNA ligation:U6-26-F: 5’TGTCCCAGGATTAGAATGATTAGGC3’dT4-R: 5’AAACGTAATATTAAACGGATGGCC3’

From the PCR-confirmed clones, plasmid DNA was isolated with a GeneJET Plasmid Miniprep (Thermo Fisher Scientific) kit and, before transformation into Agrobacterium (GV3101), the clones were confirmed through sequencing.

### 4.3. Confirmation of Clone by Colony PCR and Restriction Digestion

After transformation, cloning of sgRNA into plant expression vector pHSE401 was confirmed with colony PCR using the same primers (U6-26-F and dT4-R). sgRNAs were hybridized in a thermal cycler and cloned at BsaI restriction site in pHSE401 (https://www.addgene.org/search/catalog/plasmids/?q=pHSE401 accessed on 20 May 2018). sgRNA cloning in pHSE401 was first confirmed through colony PCR (Figure 1) with specific primers. Colony PCR products were resolved on agarose gel electrophoresis to observe required band of 280-bp. All sgRNAs were cloned in the same vector under individual promoters.

### 4.4. Extraction and Restriction Digestion of Plasmid

For plasmid extraction, a GeneJET Plasmid Miniprep (Thermo Fisher Scientific) kit was used following the kit protocol. The *E. coli* culture was grown in 10 mL of Luria–Bertani (LB) media by inoculating a colony of *E. coli* into the media. Double-distilled water (ddH2O), buffer, plasmid, and enzyme were added in an Eppendorf tube and then placed for 20 min in a water bath at 37 °C. Digestion was performed and then run on a gel of 1.2% agarose. Briefly, after colony PCR, cloned sgRNAs were also confirmed through restriction analysis. For restriction analysis of pHSE401 plasmid containing sgRNAs, a vector was digested with *Hind*III, thus producing a fragment of ~2 kb (Figure 2). *Hind*III-digested plasmid products were resolved on agarose gel electrophoresis and a fragment of ~2 kb was confirmed in vector, indicating the presence of gRNAs in the vector. Plasmid vectors which were confirmed through restriction analysis were sequenced before proceeding to transformation. Sequence analysis was obtained from Eurofins Genomics. After confirmation with sequencing, vector was transformed into Agrobacterium (strain EHA105). The map of the vector is given in Appendix A.

### 4.5. Growth Condition of Cotton Plants

In cotton growing season, cotton seeds were planted in peat moss in pots kept in dark at 28 °C. A total of 30–35 pots were maintained and watered daily. After eight to ten days, the plants were ready for transformation, as shown in Appendix A.

### 4.6. Virus Infectivity Assay in N. benthamiana

Infectious clones containing cotton leaf curl Kokhran virus and cotton leaf curl Multan betasatellite (CLCuKV/CLCuMuB) were kindly provided by Dr. Imran Amin, NIBGE, Faisalabad, Pakistan. By using an agroinfiltration technique, the infectivity of the infectious clones was checked in *N. benthamiana* plants. For infiltration of CLCuV, 14-day-old plants of *N. benthamiana* were used. An inoculation culture of Agrobacterium was grown at 28 °C containing antibiotics. The Agrobacterium cells were spun and resuspended in MgCl_2_ (20 mM) with acetosyringone. The infiltration medium was left on the bench for three hours. Syringe infiltration was done on the fully expanded and young leaves. For symptom development, nine-day-old, infiltrated plants were taken for observation. At 14 days post-inoculation (dpi), the plant leaf samples were taken for DNA extraction and either PCR or quantitative real-time PCR (qPCR) determination of virus accumulation. For PCR and qPCR, the primers and protocol provided by Khan et al. [19] were used in the present assay.

### 4.7. Transformation of Cotton

Cotton transformation was performed using hypocotyls. For transformation, Agrobacterium (strain EHA105) harboring pHSE401/pKSE401 with sgRNAs/multiple gRNAs was grown in 40 mL of LB media at 28 °C with a specific antibiotic. For the collection of hypocotyls, one-week-old cotton plants were used and sterilized with HgCl_2_ and ddH_2_O. Hypocotyls were trimmed to ~8–10 cm with a sharp sterile blade, forming blunt ends. Pellet was collected from Agrobacterium culture and reconstituted in MGL medium (5 g/L tryptone, 5 g/L NaCl, 0.1 g/L MgSO_4_, 0.25 g/L KH_2_PO_4_ and 1 g/L glycine with pH = 5.8) with subsequent addition of acetosyringone followed by shaking at 28 °C for 40 min. Sterilized hypocotyls were sliced into 1-cm pieces aseptically using an autoclaved razor and immersed in the solution for 45 min. After infection, hypocotyls were dried on filter paper and transferred to non-selective (without antibiotics) media for two days at 21 °C in the dark. Finally, hypocotyls were transferred to a callus-inducing medium (CIM) (4.33 g Murashige and Skoog (MS) minimal organics medium, 1 mL/L MS vitamin stock, 3% (*w*/*v*) glucose, 0.4 µM 2,4-D, 0.4 µM kinetin, 1000 mg/L MgCl_2_, 50 mg/L kanamycin, 400 mg/L cefotaxime, and 0.2% (*w*/*v*) Gelzan™, pH 5.8) with kanamycin or hygromycin for selection of transformants. Hypocotyls were cultured on the same media at 28 °C with a 16/8-h photoperiod. Growing calli were subsequently transferred to fresh CIM plates with antibiotics. Calli were separated from hypocotyl and shifted to regeneration media (4.33 g MS minimal organics medium, 1 mL/L MS vitamin stock, 3% glucose, 1000 mg/L MgCl_2_, 0.22% (*w*/*v*) KNO_3_, 50 mg/L kanamycin/hygromycin) and kept at 28 °C with a 16/8-h photoperiod for 21 days. For further embryogenesis, it was further sub-cultured on the media (4.33 g MS minimal organics medium, 1 mL/L MS vitamin stock, 3% glucose, 1000 mg/L MgCl_2_, 1000 mg/L glutamine, 500 mg/L asparagine, 1.25 mg/L CuSO_4_, 100 mg/L ascorbic acid, 2000 mg/L activated charcoal, 50 mg/L kanamycin/hygromycin) and kept in dark at 28 °C for 21 days. For producing embryos and transgenic plants, various methods were used [39,40,41]. Embryos were obtained in 5–6 subcultures. After elongation and rooting in 3–4 weeks, plantlets were hardened in pots. The expression of the gRNAs and Cas9 in the plantlets was confirmed with reverse transcription PCR (RT-PCR).

### 4.8. PCR-Based Confirmation of Transgene

The transgene was confirmed by PCR amplification of the gRNA. Specific primers according to the construct were used to confirm transgenic callus of cotton. The primers used for PCR confirmation are mentioned in Section 4.2.2.

In addition to cotton transformation with CRISPR/Cas constructs for CLCuV, we used DsRed as a visible marker for cotton transformation. Use of a visible marker helps to confirm and optimize transformation in the growing calli.

## 5. Conclusions

The multiplex CRISPR/Cas9 technology can be used to suppress begomoviruses in cotton. The DNA-A has a common region sequence around 200 bp which was found to be useful for viral interference while using sgRNA. Effectively designed sgRNA expression targeting non-coding and coding regions of the CLCuV genome simultaneously will improve CLCuV resistance in cotton plants. The technology may be exploited for editing other *Geminivirdae* genomes and offer an opportunity for more sustainable agricultural production systems.

## Figures and Tables

**Figure 1 ijms-22-12543-f001:**
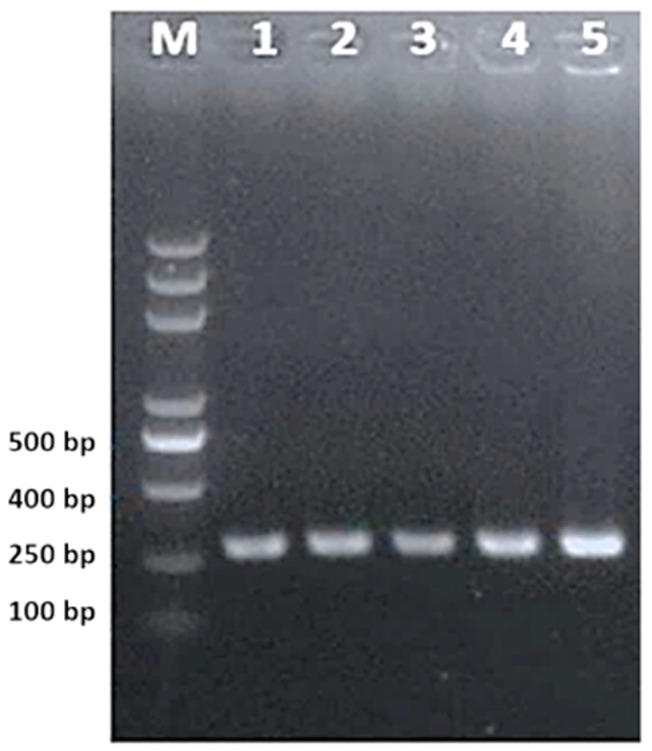
The PCR product was run on 1.2% agarose gel. Lane M is showing DNA ladder. Lanes 1–5 contain predictable clones with amplification of approximately 280 bp. M is a 500 bp DNA ladder (CWBio, China).

**Figure 2 ijms-22-12543-f002:**
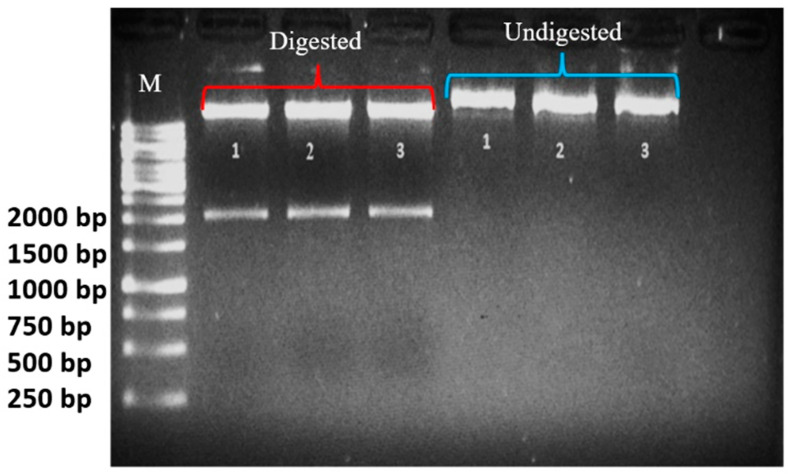
Confirmation of CRISPR construct through restriction digestion. The digested (ddH2O 15 μL + buffer 2 μL + plasmid 2 μL + HindIII 1 μL + 4 μL dye = 24 μL run) and undigested (1 μL plasmid + 4 μL ddH2O + 1 μL dye = 6 μL) reactions were loaded in 1% agarose gel. The digested band size is ~2000 bp. M is 1 kb DNA ladder (Thermo Fisher Scientific, Waltham, MA, USA).

**Figure 3 ijms-22-12543-f003:**
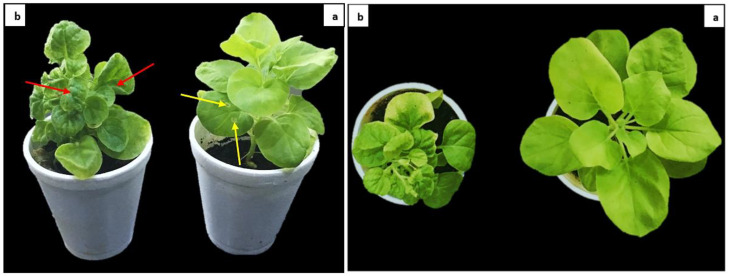
Virus infectivity assay. (a) control plants without virus infiltrated with mock vector; (b) symptoms of virus in infected plant with infectious clones. Yellow arrows show infiltrated leaf with mock vector. Red arrows show symptoms of viral infection in the leaves of infected plants infiltrated with infectious clones.

**Figure 4 ijms-22-12543-f004:**
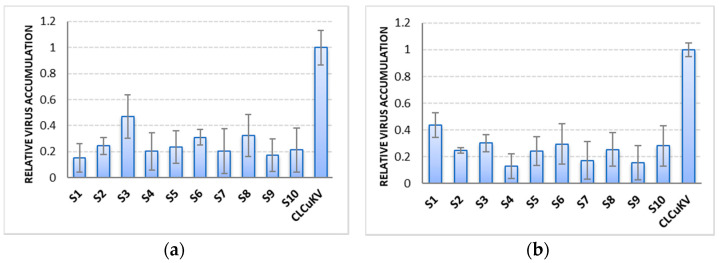
The graph bars represent a relative accumulation of CLCuKV in plants. (**a**) *N. benthamiana* co-infiltrated with virus and CRISPR; (**b**) cotton plants co-infiltrated with virus and CRISPR reagents. Bars 1–10 show low viral accumulation in the samples co-infiltrated with Cas9-gRNA, compared with reference CLCuKV.

**Figure 5 ijms-22-12543-f005:**
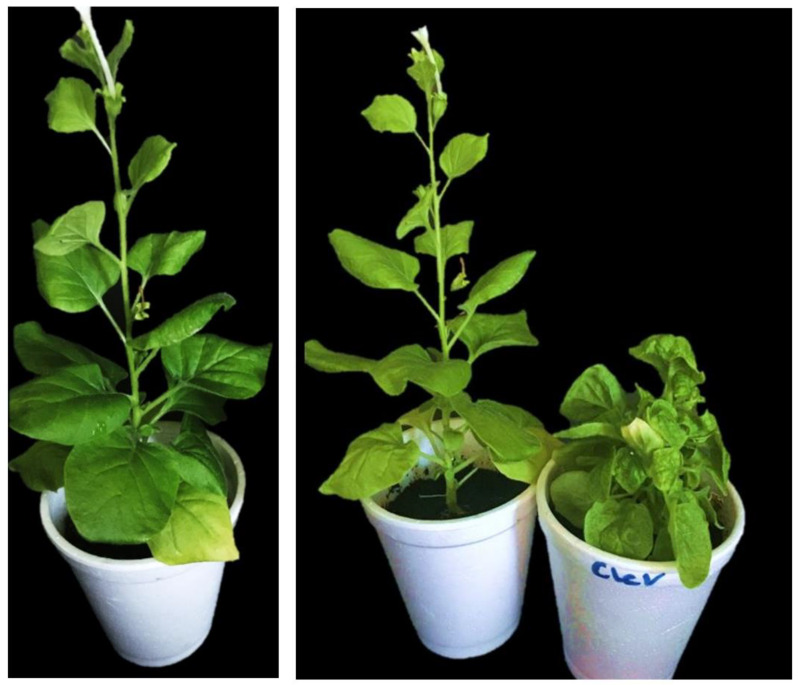
*N. benthamiana* plants showing resistance and susceptibility against CLCuV infection. The picture was taken at 30 dpi. A non-inoculated plant in 1st picture (L). In 2nd picture (R), healthy plant (**left**) (co-infiltrated) showing resistance against CLCuV at 30 dpi, while control plant (**right**) (infiltrated with virus only) is showing severe symptoms of CLCuD.

**Figure 6 ijms-22-12543-f006:**
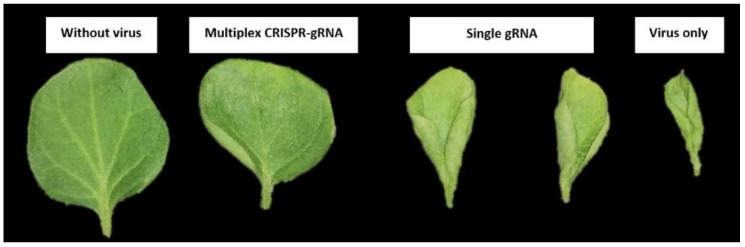
Evaluation of CLCuV resistance in *N. benthamiana* plants. Fourteen-day-old plants were infiltration with infectious clones of CLCuV. Plant with virus only and without virus were kept as controls to compare with co-infiltrated plants with CRISPR/Cas9 + CLCuV. The differences in severity of symptoms were observed.

**Figure 7 ijms-22-12543-f007:**
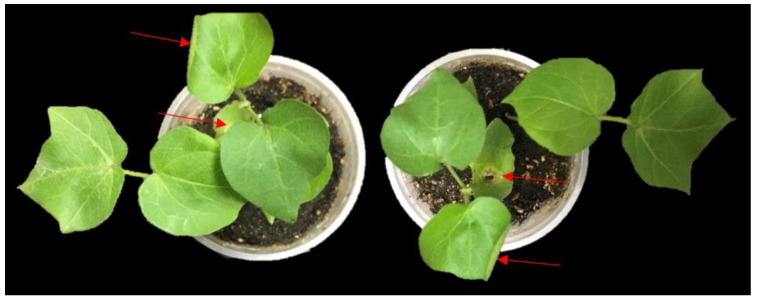
Infiltration of CLCuV in cotton seedlings. Infectious clones of CLCuV were infiltrated in 2-week-old cotton seedlings. Photographs were taken at 14–21 dpi. Mild symptoms of virus were observed in the cotton seedlings at 16 dpi.

**Figure 8 ijms-22-12543-f008:**
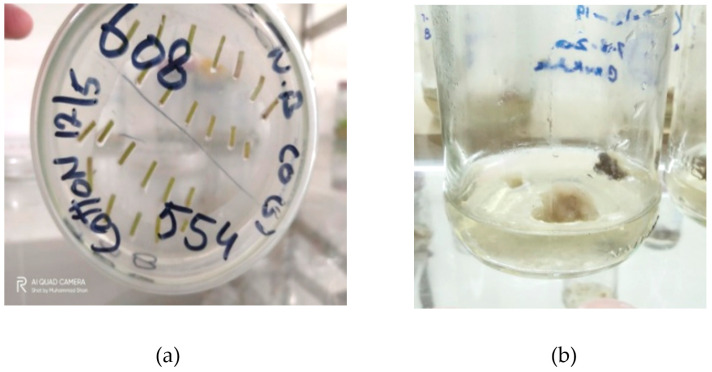
Transformed callus of cotton. (**a**) stem cuttings of cotton plant for callus induction; (**b**) mature callus was formed after six weeks.

**Figure 9 ijms-22-12543-f009:**
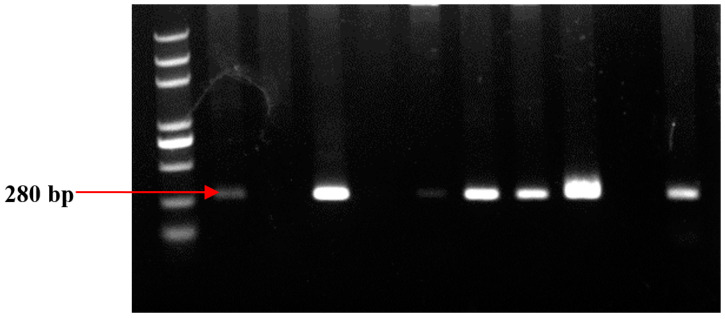
Confirmation of the transgene in callus. The expected band of 280 bp was observed in different growing calli. PCR product was run on 1.2% agarose gel.

**Figure 10 ijms-22-12543-f010:**
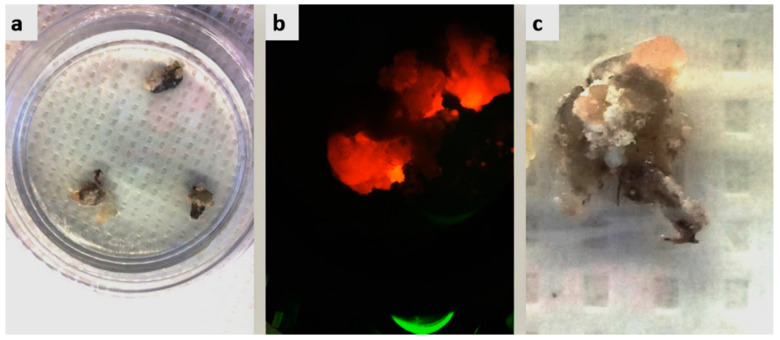
Cotton transformation optimization. DS-Red was used to optimize cotton transformation through agrobacterium-mediated transformation. (**a**) callus inducing hypocotyls are shown; (**b**) callus under fluorescence stereo-microscope showing a red color in the cells due to the expression of the DS-Red marker gene; (**c**) callus induction in the hypocotyl. Red pigment may be seen with the naked eye as well.

**Figure 11 ijms-22-12543-f011:**
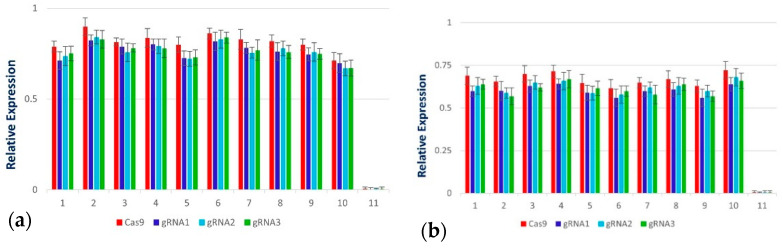
Relative expression of Cas9 and gRNAs in (**a**) *N. benthamiana* and (**b**) cotton. RNA was extracted at 2 dpi and was subjected to prepare cDNA for analyzing relative expression through RT-PCR. Adequate gene expression was observed compared to control plants.

**Figure 12 ijms-22-12543-f012:**
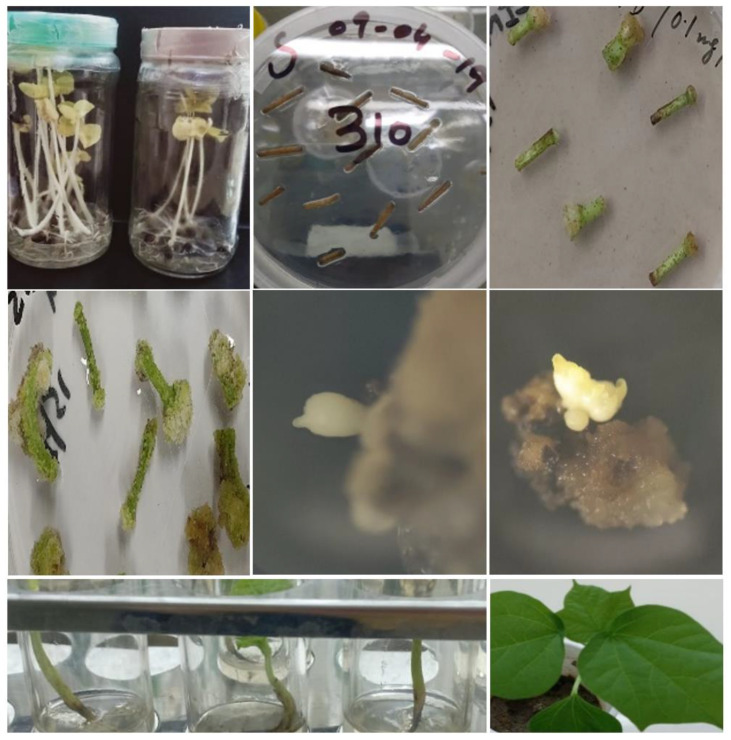
Cotton transformation. Major steps of cotton transformation have been shown in the picture starting from seedlings (left to right), hypocotyls, callus induction, growing embryos, plantlets, and transgenic plant.

**Figure 13 ijms-22-12543-f013:**
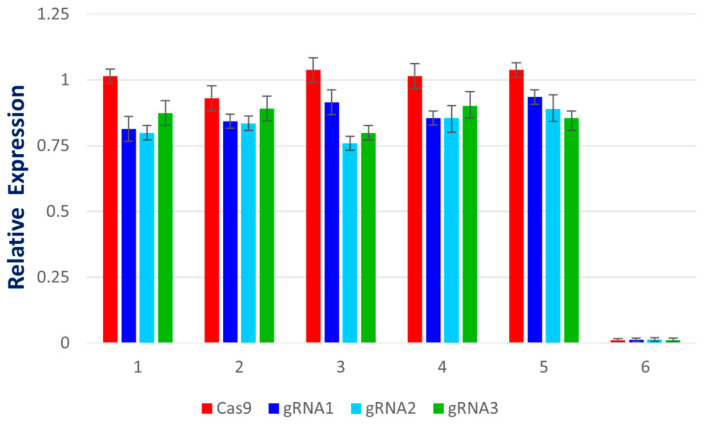
Expression of Cas9 and gRNAs in the transgenic plants. Expression of Cas9 and gRNA was quantified through RT-PCR. It was observed that all transgenic plants showed a substantial amount of expression in the leaves compared to the expression in control plants.

**Figure 14 ijms-22-12543-f014:**
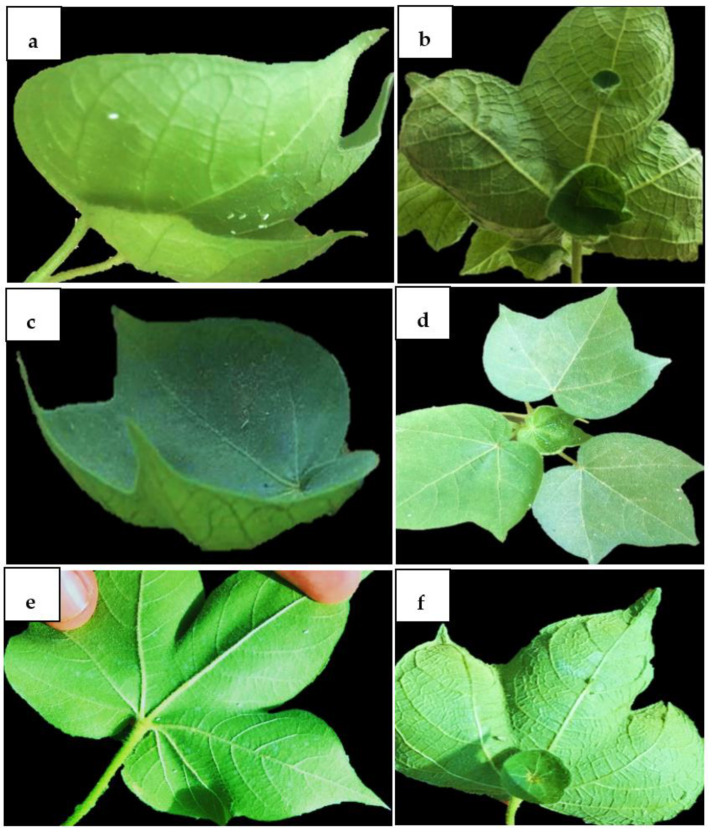
CLCuD resistance in cotton. (**a**) cotton plants were infected with viruliferous whitefly. Plant leaves showing various symptoms such as (**b**) enations, vein thickening, and (**c**) leaf curling. (**d**) CRISPR plants showed minute curling at an early stage, but no symptoms were observed at later stages such as vein thickening and enations. (**e**,**f**) showing comparison of non-transformed and transformed plants for CLCuD symptoms development at later stages.

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
