# Peer review of "Using Multiplexed CRISPR/Cas9 for Suppression of Cotton Leaf Curl Virus"

_ijms, 2021, doi:10.3390/ijms222212543_

Round 1

Reviewer 1 Report

This manuscript gives information on suppression of cotton leaf curl virus using CRISPR/Cas9. But the study is insufficient for publication, and there is a big problem with the structure of this report. I think that it is unacceptable for publication.

  1. Transgenic cotton

Authors don’t show resistance to cotton leaf curl virus using transgenic plants, though they describe production of transgenic cotton. They should add the results of tests using transgenic cotton. 

  1. Structure of the report

    Authors need to consider the structure the report.

    (1) There are unnecessary description in Result section.

      2.1 Confirmation of gRNAs and sgRNAs in Cas9 vector through colony PCR

         →simplified

      2.2 confirmation of sgRNAs through restriction analysis

         →deleted

       2.5-2.7

          →moved to Material and Methods

     (2) Material and Methods

       4.6 Virus infectivity assay in N. benthamiana

           →Authors should structure of infectious clone.

Reviewer 2 Report

The manuscript describes a CRISPR/Cas mediated genome editing against CLCuD and providing a novel stratagy for controling virus disease. However, the data of transformed cotton is not shown. Plsease also see attached file for more comments.

Round 2

Reviewer 1 Report

This manuscript gives information on suppression of cotton leaf curl virus using CRISPR/Cas9. But it has some problems. I think that it requires major revision for publication.

  1. Results

  Authors should move “2.1 Confirmation gRNAs and sgRNAs in Cas9 vector through colony PCR”, “2.2 Confirmation of sgRNAs through Restriction analysis” and “2.3 Virus infectivity assay in N. benthamiana plants” to Material and Methods.

  1. Figure 5

  Authors should add “healthy plant” that was not infected with CLCuV.

  1. Transgenic cotton

Authors should show the data about resistance of transgenic cotton. (They show only pictures in Figure S4.) If they can’t show resistance of transgenic cotton, they should delete “2.5 Transformed callus of cotton”, “2.6 PCR-based confirmation of transgene”, 2.7 Regeneration of cotton callus” and “2.8 Evaluation and screening”.

Author Response

Authors are thankful to the reviewer by providing comments for the improvement of the manuscript. All the changes suggested by Reviewer have been successfully incorporated in the text. Unnecessary descriptions have been removed. Changes have been made in Material and Methods and Results sections. Section 2.1, 2.2 and 2.3 have been moved from Results to Material and Methods under section 4.3, 4.4 and 4.6 respectively.

Note: The results of sections 2.1, 2.2 and 2.3 are given under same headings.

Authors are grateful to the Reviewer for providing his valuable suggestions. Picture of healthy non-infected plant has been added as suggested by the respected Reviewer.

Authors are very thankful to the Reviewer for reviewing their manuscript for improvements. The asked data has been provided in the revised manuscript (Section: 2.8, Figure 14). The authors, once again, want to express their gratitude for all the suggestion and comments made by the respected Reviewer for improvements to make their manuscript able to get published in IJMS.

Reviewer 2 Report

The revised version is good for publication.

Author Response

Authors are grateful to the Reviewer for appreciating their work and recommending it for publication in IJMS.

Round 3

Reviewer 1 Report

This manuscript gives information on suppression of cotton leaf curl virus using CRISPR/Cas9. I think it is acceptable for publication.